# Study of Asphalt Behavior on Pre-Wet Aggregate Surface Based on Molecular Dynamics Simulation and Surface Energy Theory

Yaoxi Cao [1], Yanhua Wang [2], He Li [3] and Wuxing Chen [4,*]

1 Jilin Communications Polytechnic, Changchun 130026, China; cyxi@jljy.edu.cn
2 Jinan City Planning and Design Institute, Jinan 250013, China; wyh-vivid@foxmail.com
3 Guidance and Service Center for Student Employment and Entrepreneurship, Jilin University, Changchun 130026, China; a326532558@jlu.edu.cn
4 School of Mines, China University of Mining and Technology, Xuzhou 221000, China
* Correspondence: 6488@cumt.edu.cn

**Abstract:** The improvement of the performance of asphalt mixtures has been studied by scholars. This research proposes a new asphalt–mineral interface formation method, which is a pre-wet bitumen–mineral mixture. The formation process of the pre-wet asphalt–mineral interface was simulated by molecular dynamics simulation software. The diffusion coefficient, concentration distribution, and interfacial energy of the asphalt on the surface of the pre-wet mineral material and non-pre-wet mineral material were compared and analyzed. The simulation results show that the mineral surface diffusion rate of the asphalt after pre-wetting is increased by more than 50%, and the concentration in the X, Y, and Z directions is reduced by 0.8%, 4.6%, and 7.8%, respectively. At the same time, the interface energy between the bitumen and the pre-wet mineral was increased by more than 8%. The results of the molecular dynamics model are verified based on the surface energy theory and contact angle test. The experimental results show that the contact angle of the asphalt is smaller and the diffusion performance is better after pre-wetting. At the same time, the interface adhesion work between the asphalt and wet mineral surface increased by 4.3% in a dry environment, and the peeling work decreased by 41.1% in a water environment. Therefore, the author believes that the pre-wetting technology of the asphalt mixture has a certain feasibility and practicability.

**Keywords:** asphalt–mineral interface; pre-wet technology; molecular dynamics simulation; surface energy theory; interface performance analysis

## 1. Introduction

Asphalt, which has many advantages as a pavement bonding material, has become an indispensable material for highway construction [1–4]. However, many issues, such as potholes, cracks, and ruts, also appear during the use of asphalt pavements. An analysis has revealed that asphalt pavement issues are caused by the decrease in the adhesion effect of the asphalt–stone interface. Increasing the workability of the asphalt and stone in the production process of the asphalt mixture is crucial to improve the adhesion between the asphalt and stone [5–7].

The number of studies on improving the adhesion between asphalt and mineral aggregates has increased in recent years. At present, the main method to improve the performance of asphalt mixtures is the addition of modifiers. S. Hayash et al. compounded the asphalt with epoxy resin and other modifiers. Their experimental results showed that the asphalt mixture modified by the epoxy resin compound has good strength, durability, and anti-rutting capability [8–10]. Zhang et al. used montmorillonite (MMT) to modify SBR asphalt and found that SBR and MMT can form a special structure in asphalt, which can inhibit the flow of asphalt molecules and improve the resistance to permanent asphalt deformation [11]. Feng et al. introduced polyphosphoric acid (PPA) and sulfur (S) into SBR asphalt and then studied the rheological properties and microstructure characteristics

of modified asphalt. Their results showed the vulcanization and coagulation produced by PPA and S. The glue effect significantly improved the stability of the modified asphalt system and the high-temperature deformation resistance of the asphalt [12]. Sun et al. used nano-SiO2 to modify asphalt and found that the prepared nano-SiO2 modified asphalt has high- and low-temperature performance and anti-aging capability [13].

Computer simulation technology plays an increasingly important role in scientific research with the development of science and technology. In recent years, molecular dynamics simulation has gradually highlighted its advantages in the study of asphalt [14–17]. Chen, ZX et al. used molecular dynamics simulation to predict the interface characteristics of asphalt and mineral aggregates effectively [18]. Ding, YJ et al. analyzed the influence of SBS on the agglomeration behavior of asphalt binder molecules through molecular dynamics simulation software. Their analysis results revealed the susceptibility of the agglomeration structure of asphalt molecules to the influence of SBS when the side branch of the asphaltene alkane is long [19]. Qu, X et al. also utilized molecular dynamics simulation software to examine the influence of paraffin on the performance of an asphalt binder. They found that paraffin will reduce the high-temperature stability of the asphalt binder and affect its road performance [20]. Wang, H et al. simulated and analyzed the influence of moisture on the interface between asphalt and mineral aggregates through molecular dynamics simulation software and discovered that moisture reduces the adhesion between the asphalt and mineral aggregates [21]. Xu et al. [22] calculated the interaction energy and adhesion work of the asphalt–aggregate interface, and evaluated the adhesion performance of the interface in a water environment by using the bond energy parameters under wet and dry conditions. Yao et al. [23] calculated the asphalt–aggregate contact angle by the MD method to evaluate the interface wettability and explore the interface behavior between the nano-asphalt and aggregate.

In recent years, the methods used to improve the performance of asphalt binders have mainly focused on the research on asphalt modifiers. A few scholars have comprehensively studied the construction technology of asphalt mixtures. The mixing of asphalt mixtures is divided in this study into two stages. The first stage uses a high-grade asphalt to wet the minerals, and the second stage performs mixing to improve the uniformity of the asphalt mixture. In this study, the feasibility and practicability of pre-wet asphalt mixture technology was studied through molecular dynamics simulation, surface energy theory, and a contact angle test.

## 2. Materials

The base asphalt used in this research is 90# asphalt called "Panjin." Meanwhile, "Panjin" 110# asphalt was used in the first stage of pre-wetting, and "Panjin" 90# asphalt was used in the second stage of mixing. The basic technical indicators of the two asphalts are shown in Tables 1 and 2.

**Table 1.** The 90# asphalt's basic technical indicators.

| Basic Indicator | 25 °C Penetration (0.1 mm) | 25 °C Ductility (cm) | Softening Point (°C) | Flash Point (°C) | Density (g·cm$^{-3}$) |
|---|---|---|---|---|---|
| Test results | 89.6 | >100 | 48.7 | 265 | 1.003 |
| Requirements | 80–100 | >100 | 42–52 | ≥245 | - |
| Test procedure | GB/T0606-2011 [24] | GB/T0605-2011 [24] | GB/T0606-2011 [24] | GB/T0611-2011 [24] | GB/T0603-2011 [24] |

**Table 2.** The 110# asphalt's basic technical indicators.

| Basic indicator | 25 °C Penetration (0.1 mm) | 25 °C Ductility (cm) | Softening Point (°C) | Flash Point (°C) | Density (g·cm$^{-3}$) |
|---|---|---|---|---|---|
| Test results | 111.3 | >100 | 42.5 | 245 | 1.001 |
| Requirements | 100–120 | >100 | - | - | - |
| Test procedure | GB/T0606-2011 [24] | GB/T0605-2011 [24] | GB/T0606-2011 [24] | GB/T0611-2011 [24] | GB/T0603-2011 [24] |

## 3. Molecular Model Establishment and Rationality Verification

### 3.1. Force Field

When performing molecular dynamics calculations, the choice of force field is very important, because it determines the calculation method between the molecules. We must choose a suitable force field, otherwise the calculated results will have huge deviations. Similarly, the choice of the ensemble is also very important. It is necessary to choose a suitable ensemble according to the requirements of the calculation, so that reasonable results can be obtained. According to the needs of this research, the force field selected in this paper is force field COMPASS II, which is an upgraded version of force field COMPASS, which best meets the calculation requirements of this research. The ensemble selected in this study reasonably chooses an equal pressure ensemble (NPT) and canonical ensemble (NVT) according to the needs of the calculation. The simulated time step is 100 ps. The simulation time step was 1 fs, the acquisition track frequency was 1 ps, the temperature control method was Anderson, and periodic boundary conditions were adopted.

### 3.2. Establishment of Asphalt Molecular Model

The molecular composition of asphalt is extremely complex, but most of these molecules are composed of C, H, S, and N elements [25]. The asphalt molecular model used in this study was proposed by Li and Greenfield et al., because this 12-component average asphalt molecular model is currently the most commonly used [26–29]. The 12-component asphalt model established by Materials Studio 2019 in Figure 1 is used in this study.

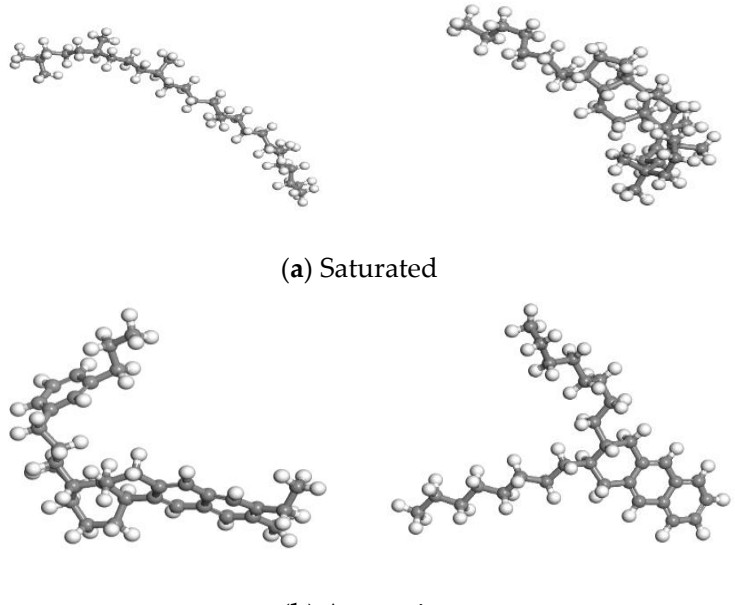

(**a**) Saturated

(**b**) Aromatic

**Figure 1.** *Cont.*

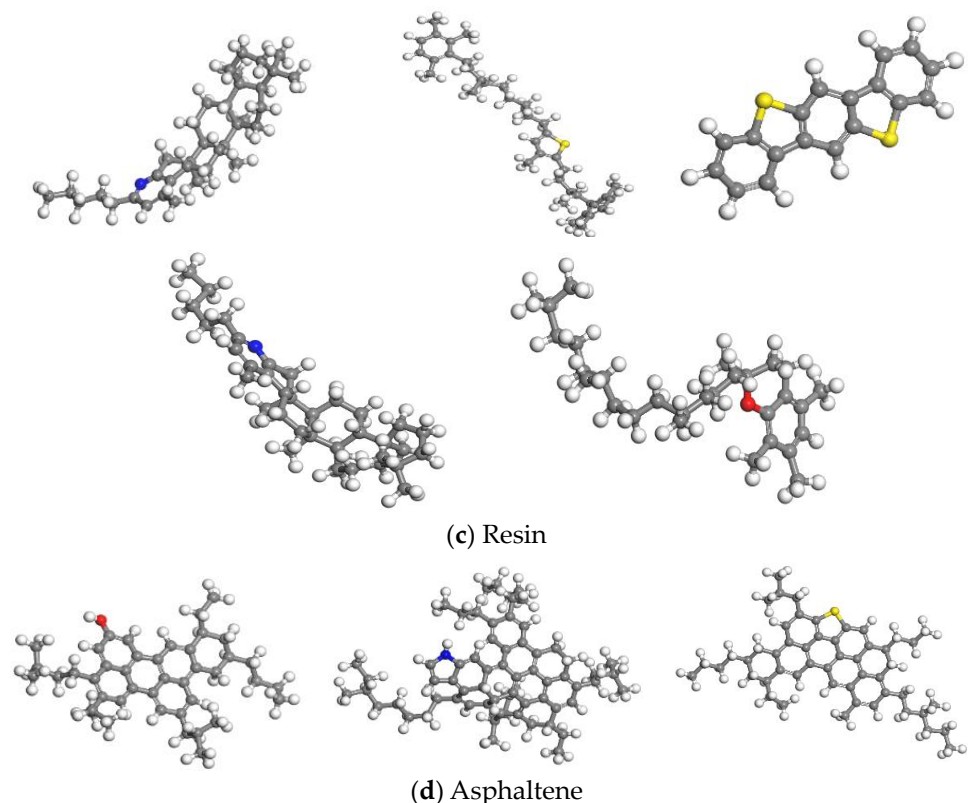

(**c**) Resin

(**d**) Asphaltene

**Figure 1.** Asphalt component model. (**a**) Saturated phenol; (**b**) aromatic; (**c**) resin molecule; (**d**) asphaltene.

Molecular simulation software is used in accordance with the proportion of each component of the asphalt as a whole to establish an asphalt molecular model. The ratio of each component of 90# and 110# asphalt is shown in Table 3.

**Table 3.** Molecular composition of asphalt.

| Asphalt Type | | 90# Asphalt | 110# Asphalt |
|---|---|---|---|
| | Saturate A | 11 | 11 |
| | Saturate B | 10 | 10 |
| | Aromatic A | 29 | 29 |
| | Aromatic B | 34 | 34 |
| | Resin A | 4 | 3 |
| **Asphalt component** | Resin B | 4 | 3 |
| | Resin C | 6 | 5 |
| | Resin D | 4 | 3 |
| | Resin E | 5 | 4 |
| | Asphaltene A | 5 | 4 |
| | Asphaltene B | 3 | 2 |
| | Asphaltene C | 4 | 3 |

The molecular simulation software Materials Studio 2019 was used in accordance with the number of molecules in Table 3 to build the 90# and 110# asphalt models. The 110# and 90# asphalts were, respectively, used for the first-stage pre-wetting and second-stage mixing. The "Amorphous" module in Materials Studio 2019 was utilized to build the asphalt molecular models. The asphalt molecular model must be structurally optimized to stabilize its internal structure and annealed under the regular ensemble (NVT) to balance its internal energy. The asphalt molecular model is shown in Figure 2.

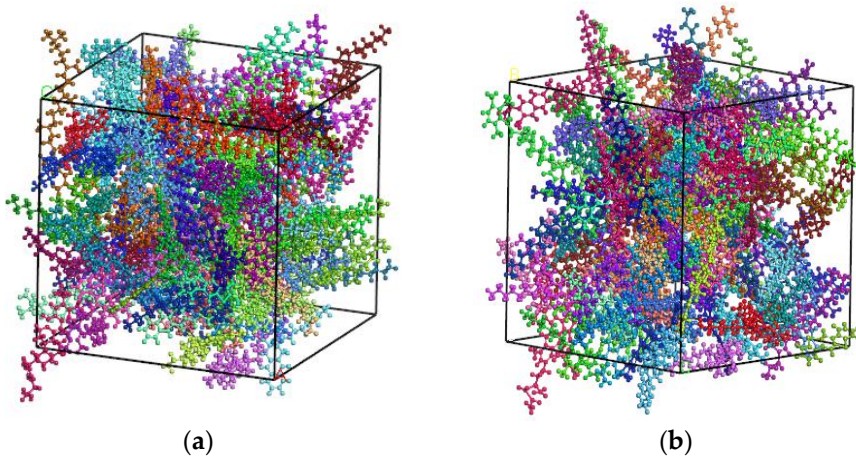

**Figure 2.** Asphalt molecular model. (**a**) 90# Asphalt, (**b**) 110# asphalt.

*3.3. Establishment of SiO₂ Crystal Model*

Among the mineral materials used in road engineering, $SiO_2$ is the main component of the mineral materials used [30]. Therefore, this study uses $SiO_2$ as the surface model of the mineral material. The crystal bond length and bond angle of $SiO_2$ are a = 4.913 Å, b = 4.913 Å, c = 5.405 Å, $\alpha$ = 90°, $\beta$ = 90°, and $\gamma$ = 120°. The single unit cell is established through the "Crystals" module of the molecular simulation software according to the unit cell data of $SiO_2$, and then the single unit cell slice is copied and added to the vacuum layer to create the mineral surface model. The established $SiO_2$ crystal model is shown in Figure 3.

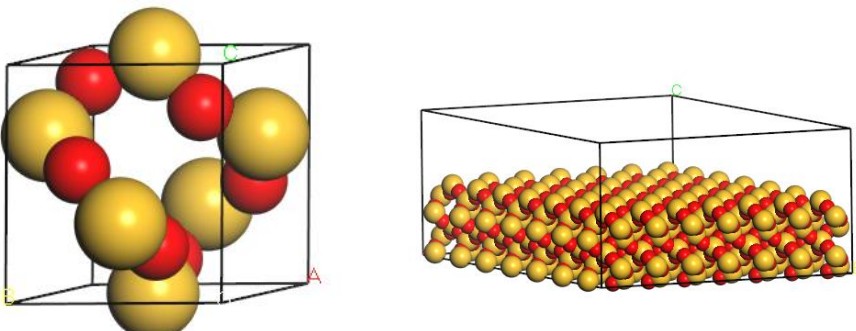

**Figure 3.** SiO₂ crystal model.

*3.4. Verification of the Rationality of the Asphalt Molecular Model*

3.4.1. Density

Verifying the error between the density of the established asphalt model and the actual asphalt density is an effective means to test whether the asphalt model is reasonable. There is definitely an error between the density of the asphalt model and the actual asphalt density, but as long as the density of the asphalt model can represent the actual asphalt, it is considered reasonable. The ratio of the asphalt model density to the actual asphalt density is shown in Table 4.

**Table 4.** Asphalt model density and measured density.

| Asphalt Type | Model Density (g/cm³) | Measured Density (g/cm³) | Density Ratio (%) |
|---|---|---|---|
| 90# asphalt | 1.013 | 1.036 | 97.8 |
| 110# asphalt | 1.011 | 1.028 | 98.3 |

As shown in the data in Table 4, the ratio of the density of the two asphalt models to the actual asphalt density is above 97%, and the asphalt model is considered to represent the actual asphalt.

### 3.4.2. Solubility Parameters

The solubility parameter is the square root of the cohesive energy density of the material, and there is a reasonable range for the solubility parameter, namely 15.3 $(J/cm^3)^{1/2}$–23 $(J/cm^3)^{1/2}$. If the solubility parameter is within this reasonable range, it is considered to be blendable as a whole. On the contrary, if the parameter is not in the 15.3 $(J/cm^3)^{1/2}$–23 $(J/cm^3)^{1/2}$ interval, then mixing the liquid as a whole is difficult.

According to the data in Table 5, one of the solubility parameters of the two asphalts is 17.291 $(J/cm^3)^{1/2}$ and the other is 18.262 $(J/cm^3)^{1/2}$, both of which are in the middle of the required range. Through the verification of the density and solubility parameters of the asphalt model, the author believes that the established asphalt model can represent the actual asphalt for subsequent research.

**Table 5.** Cohesive energy density and solubility parameters.

| Asphalt | Cohesive Energy Density/$(J/m^3)$ | Solubility Parameter/$(J/cm^3)^{1/2}$ | Electrostatic Solubility Parameter/$(J/cm^3)^{1/2}$ | Van der Waals Solubility Parameter/$(J/cm^3)^{1/2}$ |
|---|---|---|---|---|
| 90# asphalt | $2.911 \times 10^8$ | 17.291 | 1.509 | 16.786 |
| 110# asphalt | $3.335 \times 10^8$ | 18.262 | 1.072 | 17.764 |

## 4. Molecular Dynamics Simulation of Asphalt Mixture Pre-Wetting Technology

### 4.1. Establishment of Asphalt–Mineral Material Pre-Wet Interface Model

Two asphalt models are established in accordance with the proportions of the various components of asphalt as shown in Table 3, and the "Build Layers" module is used to assemble the established asphalt and mineral material models into an asphalt–mineral material interface model. The asphalt–mineral material model without the pre-wetting technology is divided into three layers: the first layer is $SiO_2$, the second layer is 90# asphalt, and the third layer is a 30 Å vacuum layer. The asphalt–mineral interface model after pre-wetting is divided into four layers: the first layer is $SiO_2$, the second layer is 110# asphalt used for pre-wetting, the third layer is 90# asphalt used for secondary mixing, and the last layer is a 30 Å vacuum layer. The established asphalt–mineral material model is shown in Figure 4.

### 4.2. Analysis of Interface Adhesion of Two Asphalt–Mineral Models

The size of the interface energy can reflect the adhesion effect of the interface. Two asphalt–mineral interface models were used in this study to simulate the molecular dynamics at 25 °C and 165 °C, and the interface energy was calculated. Equation (1) is the formula used to calculate the interface energy. Table 6 records the calculation results of the interface energy of the two asphalt–mineral interfaces at different temperatures, and the data in Table 6 are plotted in Figure 5.

$$E_{Interface} = (E_{Dynamics} - E_{Anneal})$$ (1)

where $E_{Interface}$ is the interface energy, $E_{Dynamics}$ is the model energy after dynamic simulation, and $E_{Anneal}$ is the model energy after annealing.

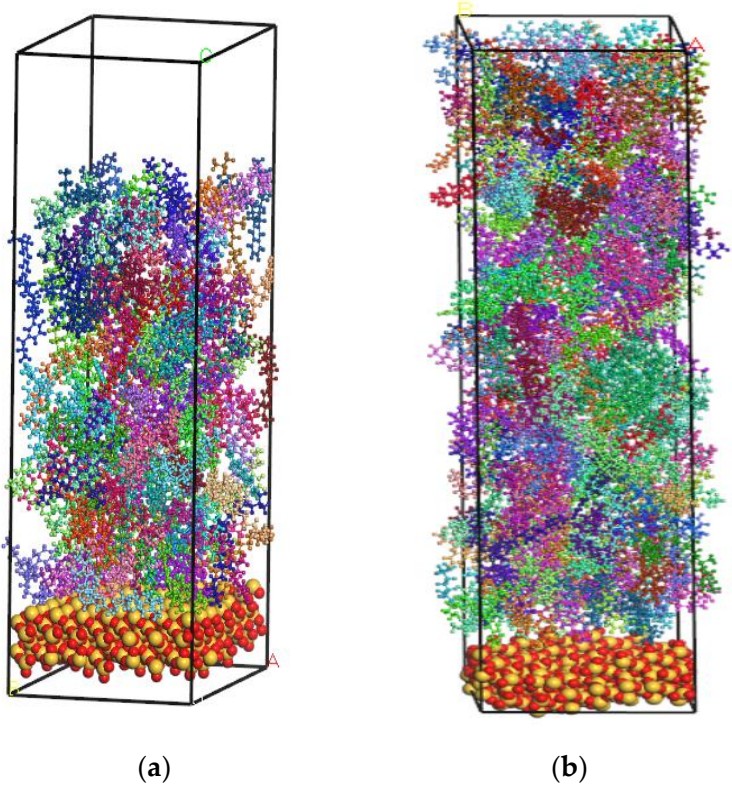

(**a**)　　　　　(**b**)

**Figure 4.** (**a**) Un-pre-wetted asphalt–mineral interface model; (**b**) asphalt–mineral interface model after pre-wet treatment.

**Table 6.** Two kinds of asphalt–mineral material model interface energy.

| Temperature (°C) | Asphalt | $E_{Dynamics}$ (kcal/mol) | $E_{Anneal}$ (kcal/mol) | $E_{Interface}$ (kcal/mol) |
|---|---|---|---|---|
| 25 | Unpre-wet | 11,072.487944 | 41,880.716774 | −30,808.22883 |
| | Pre-wet | 4036.826297 | 37,642.139897 | −33,605.31360 |
| 165 | Unpre-wet | 6953.499457 | 35,892.273840 | −28,938.774383 |
| | Pre-wet | 3128.983748 | 34,984.372845 | −31,855.389097 |

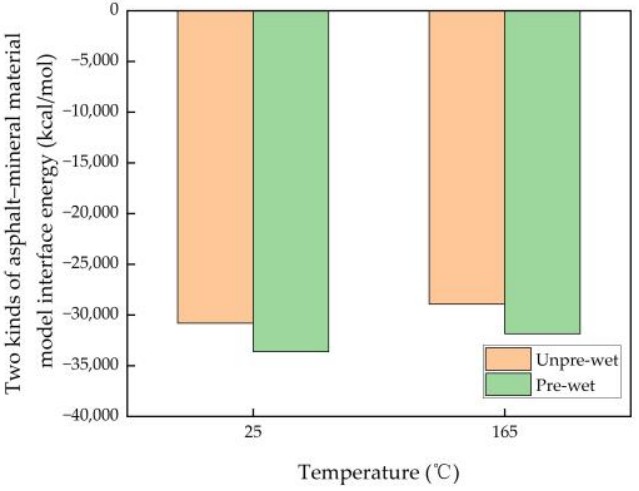

**Figure 5.** Two kinds of asphalt–mineral material model interface energy.

The data in Table 6 and Figure 5 show that the asphalt–mineral interface with the pre-wetting technology has substantial adhesion qualities whether at 25 °C or 165 °C. This result indicates that the asphalt mixture pre-wetting technology can effectively improve the adhesion between the asphalt and the mineral aggregate and enhance the performance of the asphalt mixture. The surface of the asphalt and the mineral material will have an enhanced contact effect after adopting the pre-wetting technology of the asphalt mixture. Therefore, the asphalt and the mineral material are mixed uniformly, thereby improving the integrity of the asphalt mixture. The data in Table 6 show that the adhesion work of the asphalt–mineral surface using the asphalt mixture pre-wetting technology can be increased by approximately 8%–10%. The theory of surface energy indicates that the light components are crucial in the penetration of asphalt to the mineral surface during the formation of the asphalt–mineral interface. Therefore, the surface of the mineral material after the 110# bitumen pre-wetting has substantial light components, facilitating the full contact of the asphalt–mineral material and forming a firm interface.

*4.3. Analysis of the Diffusion Rate of Asphalt on the Surface of Mineral Aggregates*

4.3.1. Mean Square Displacement

The particle is in a state of constant motion, and the mean square displacement (MSD) is the average square value of the displacement when the object moves to time t. The MSD can characterize the speed of asphalt diffusion on the surface of the mineral aggregates. The MSD of the two asphalts on the surface of the aggregate was calculated in this study at 25 °C and 165 °C. The MSD calculation equation is shown in Equation (2), and the calculation results are presented in Figure 6.

$$MSD(t) = \frac{1}{N}\sum_{i=1}^{N}[r_i(t) - r_i(0)]^2 \tag{2}$$

where $r_i(t)$ is the displacement of the particle at time t, $r_i(0)$ is the displacement of the particle at the initial time, and $N$ is the number of particles.

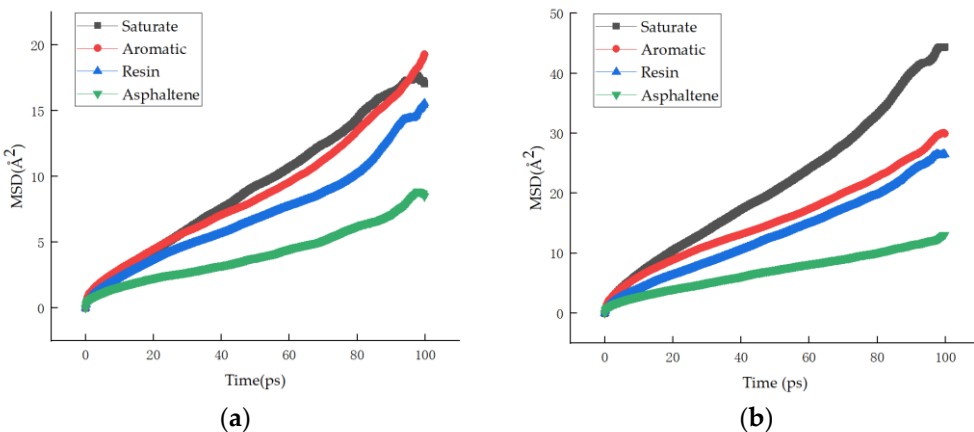

**Figure 6.** Asphalt four-component MSD. (**a**) 25 °C; (**b**) 165 °C.

The particle diffusion coefficient is characterized by the slope of the MSD curve. The particle diffusion speed is fast when the slope of the MSD curve is large. The particle diffusion coefficient calculation formula is shown in Equation (3).

$$D = \frac{1}{6}\lim\frac{d}{dt}\sum_{i=1}^{N}[r_i(t) - r_i(0)]^2 \tag{3}$$

If the slope of the curve in Equation (3) is approximated as the MSD in Equation (2), then the calculation formula for the diffusion coefficient D of the particles is shown in Equation (4).

$$D = \frac{A}{6} \tag{4}$$

### 4.3.2. Analysis of the Diffusion Law of the Four Components of Asphalt on the Surface of Mineral Aggregates

In this study, the MSD of the four components of the asphalt on the surface of the mineral materials at 25 °C and 165 °C was first calculated, as shown in Figure 6. Understanding the diffusion law of the four components of the asphalt on the surface of the mineral materials can facilitate learning and reveal the law of asphalt diffusion on the surface of minerals.

Figure 6 shows that the diffusion rate of the saturated and aromatic components on the surface of the mineral aggregates is significantly higher than that of the colloids and asphaltenes whether at 25 °C or 165 °C. Meanwhile, the diffusion rate of asphaltene on the surface of the mineral aggregate is the slowest. This rule is mainly due to the saturated and aromatic components, which are both light components. Thus, the saturated components have the smallest molecular weight, gums and asphaltenes are heavy components, and asphaltenes have the largest molecular weight. The light component is the condition in which the asphalt can quickly diffuse on the surface of the mineral material. Therefore, the diffusion rate on the surface of the mineral material is fast when the molecular weight of the component is small.

### 4.3.3. Analysis of the Law of Diffusion of Asphalt on the Surface of Pre-Wet Mineral Aggregates

The author uses molecular dynamics simulation software to simulate the diffusion rates of asphalt on the surface of pre-wet and non-pre-wet minerals. The simulation result is shown in Figure 7.

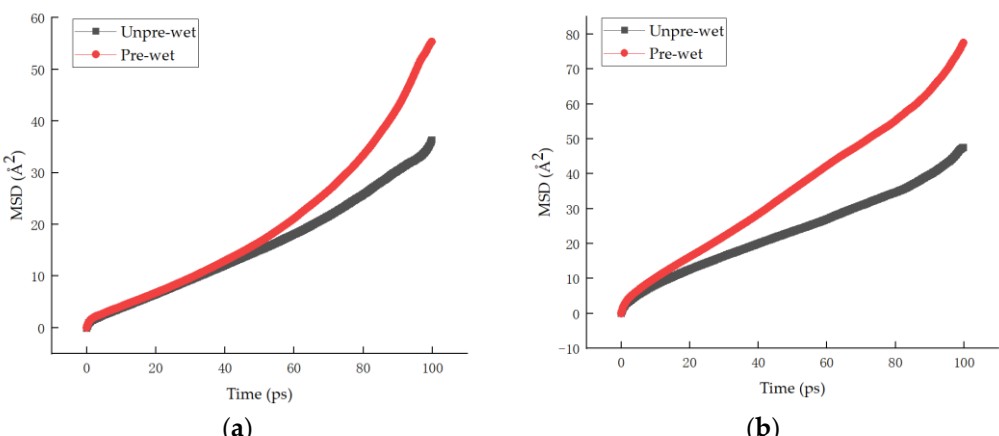

**Figure 7.** MSD of asphalt on the surface of aggregate. (**a**) 25 °C; (**b**) 165 °C.

The data analysis in Figure 7 reveals that the MSD of the asphalt on the surface of the pre-wetted mineral material is increased by approximately 52% at a temperature of 25 °C compared with that of the non-pre-wet mineral material surface. By contrast, the MSD of the asphalt on the surface of the pre-wetted mineral aggregate is approximately 62% higher than that on the surface of the non-pre-wetted aggregate at a temperature of 165 °C.

The formation process of a strong asphalt mixture affects the diffusion process of the asphalt on the mineral material surface. A strong asphalt–mineral material interface and an improved performance of the entire asphalt mixture lead to a satisfactory diffusion effect. Asphalt has a fast diffusion rate on the surface of the pre-wet mineral material. Therefore,

the pre-wet technology of the asphalt mixture can effectively improve the performance of the asphalt mixture.

The formation of the interface between the asphalt and mineral materials can be divided into two processes. First, the asphalt wets the surface of the mineral aggregate, and then the interface is formed. The wetting effect of the asphalt on the surface of the mineral aggregate is crucial to the performance of the asphalt–mineral interface after it is formed. According to this study, the pre-wet asphalt mixture technology can effectively increase the diffusion rate of the asphalt on the surface of the mineral aggregate, and by calculating the asphalt–mineral interface energy, the feasibility of using the asphalt mixture is confirmed.

### 4.4. Analysis of Concentration Distribution of Asphalt on Mineral Surface

The formation of the interface between the asphalt and ore surface is the process of the asphalt gradually coating ore. The concentration of the asphalt in all directions on the ore surface can analyze the uniformity of the asphalt dispersion on the ore surface. The concentration distribution of the two types of bitumen on the mineral surface is shown in Figure 8. The peak concentration of the asphalt in three directions and corresponding positions are shown in Table 7.

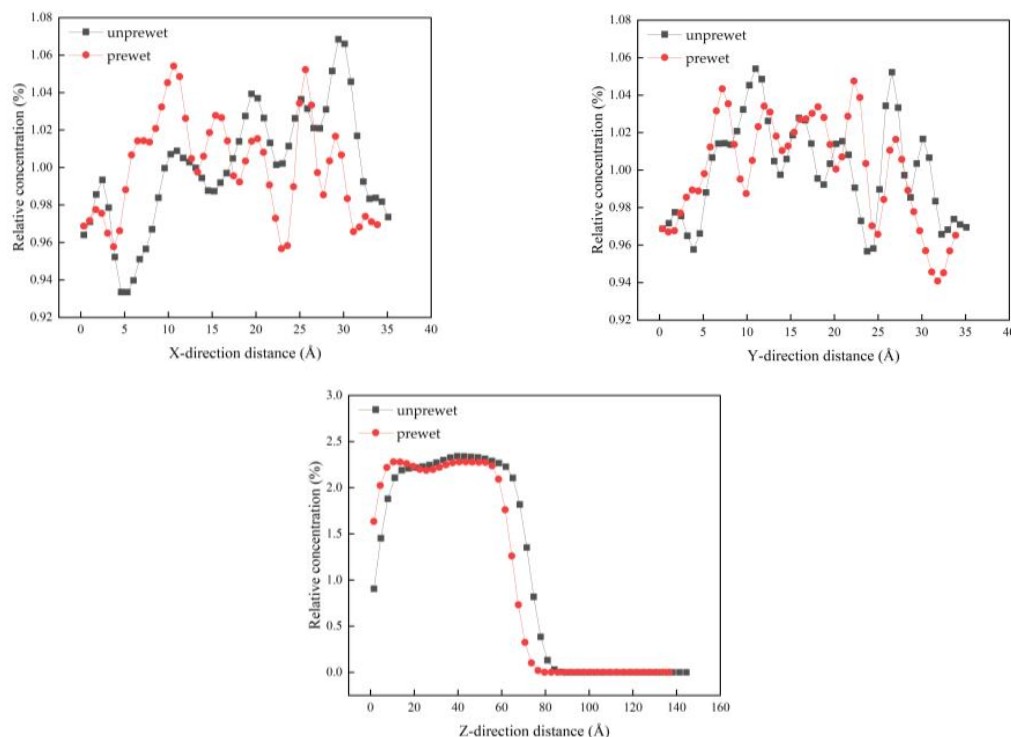

**Figure 8.** Concentration distribution.

**Table 7.** Peak concentration and corresponding position.

| Direction | Un-Pre-Wet | Pre-Wet |
|:---:|:---:|:---:|
| X | 1.063% | 1.054% |
| | 3.91 Å | 10.61 Å |
| Y | 1.098% | 1.047% |
| | 14.59 Å | 19.50 Å |
| Z | 2.472% | 2.280% |
| | 34.84 Å | 10.52 Å |

Through the analysis of the data in Figure 8 and Table 7, it can be seen that the concentration of the asphalt in the X direction and Y direction on the mineral surface is

about 1%, and the concentration in the Z direction is about 2.5 times that in the X direction and Y direction. After pre-wetting, the concentration of the asphalt decreases somewhat. After pre-wetting the 110# asphalt, the concentration in the X direction decreased by 0.8%, the concentration in the Y direction decreased by 4.6%, and the concentration in the Z direction decreased by 7.8%. This shows that after pre-wetting, the asphalt is more evenly distributed on the mineral surface, which is more favorable for the formation of the interface between the asphalt and mineral surface.

### 4.5. Feasibility Analysis of Pre-Wetting Technology Based on Surface Energy Theory

The adhesion process between the asphalt and mineral surface can be interpreted as the process of adsorbing the asphalt molecules on the stone surface to form an asphalt–stone interface to reduce the free energy of the stone surface. This section illustrates the feasibility of the ready-mix technology based on the surface energy theory.

#### 4.5.1. Analytical Method for Oil–Stone Interface Properties of Asphalt Mixture

The surface energy theory points out that the adhesion process of two interfaces (such as a and b) is the process in which interface a and interface b disappear to form the a and b common interface, and the surface free energy change per unit area during the adhesion process is shown in Formula (5).

$$W_{ab} = \gamma_a + \gamma_b - \gamma_{ab} \tag{5}$$

where $W_{ab}$ is the adhesion work, $\gamma_a$ is the surface energy of surface *a*, $\gamma_b$ is the surface energy of surface *b*, and $\gamma_{ab}$ is the surface energy between the interfaces.

In 1805, Young proposed the famous Young's equation, the main theory of which is that the force at the interface when a liquid is in contact with a solid has the relationship shown in Formula (6).

$$\gamma_{sg} = \gamma_{sl} + \gamma_{lg} \cos \theta \tag{6}$$

where $\gamma_{sg}$ is the interfacial tension between solid and air, $\gamma_{sl}$ is the interfacial tension between solid and liquid, $\gamma_{lg}$ is the interfacial tension between liquid and gas, and $\theta$ is the solid–liquid contact angle.

Formula (7) can be obtained from Formulas (5) and (6).

$$W_{sl} = \gamma_l(1 + \cos \theta) \tag{7}$$

Combining the van der Waals theory and Lewis acid–base theory, and ignoring the intermolecular forces, the formula of adhesion work can be calculated by the polarity and dispersion components of the two phases forming the interface, as shown in Formula (8).

$$W_{sl} = 2\sqrt{\gamma_s^d \gamma_l^d} + 2\sqrt{\gamma_s^p \gamma_l^p} \tag{8}$$

where $\gamma_s^d$ is the dispersion component of the solid surface, $\gamma_l^d$ is the dispersion component of the liquid surface, $\gamma_s^p$ is the polar component of the solid surface, and $\gamma_l^p$ is the polar component of the solid surface.

#### 4.5.2. Asphalt Surface Free Energy Calculation

The adhesion work is a parameter index to measure and determine whether the interface between the asphalt and stone is firm or not. According to Formulas (7) and (8), it is necessary to know the surface free energy and contact angle of the two substances in order to obtain the adhesion work of the two substances. Or, the adhesion work can be obtained by the dispersion and polarity components of the two surface energies, and Formula (9) can be obtained according to Formulas (7) and (8).

$$\gamma_l(1 + \cos \theta) = 2\sqrt{\gamma_s^d \gamma_l^d} + 2\sqrt{\gamma_s^p \gamma_l^p} \tag{9}$$

It can be seen from Formula (5) that the surface energy, dispersion component, and polarity component of the measured surface can be obtained by dropping two or more liquids with known surface energies on the measured surface and measuring their contact angles. Distilled water and ethylene glycol are the two probe liquids selected in this paper, and their surface energy parameters are shown in Table 8.

**Table 8.** Surface energy parameter.

| Liquid Type | $\gamma_l$ (mJ/m$^2$) | $\gamma_l^d$ (mJ/m$^2$) | $\gamma_l^p$ (mJ/m$^2$) | Polarity Type |
|---|---|---|---|---|
| Distilled water | 72.8 | 21.8 | 51.0 | bipolar |
| Ethylene glycol | 48.3 | 29.3 | 19.0 | unipolarity |

The 90# asphalt selected in this paper is prepared into an asphalt film slide according to the requirements, and then the contact angle of the 90# asphalt surface is measured with distilled water and ethylene glycol, and the surface energy is calculated. The measurement results of the contact angle and the surface energy are shown in Table 9.

**Table 9.** Contact angle and surface energy.

| Contact Angle | | Surface Energy | | |
|---|---|---|---|---|
| Distilled Water (°) | Ethylene Glycol (°) | $\gamma_a$ (mJ·m$^{-2}$) | $\gamma_a^d$ (mJ·m$^{-2}$) | $\gamma_a^p$ (mJ·m$^{-2}$) |
| 92.1 | 74.5 | 19.456 | 12.885 | 6.571 |

The contact part of the distilled water on the 90# asphalt surface is about 90°, indicating that there is a huge difference in the nature and polarity of distilled water and asphalt, and distilled water cannot infiltrate the asphalt surface.

### 4.5.3. Asphalt–Mineral Surface Contact Angle Test

The adhesion process between the asphalt and mineral surface can be interpreted as the process of adsorbing the asphalt molecules on the stone surface to form an asphalt–stone interface to reduce the free energy of the stone surface. The contact angle can reflect the wettability of the liquid to the solid surface. The contact angle experiment process is shown in Figure 9. The experimental results are shown in Table 10.

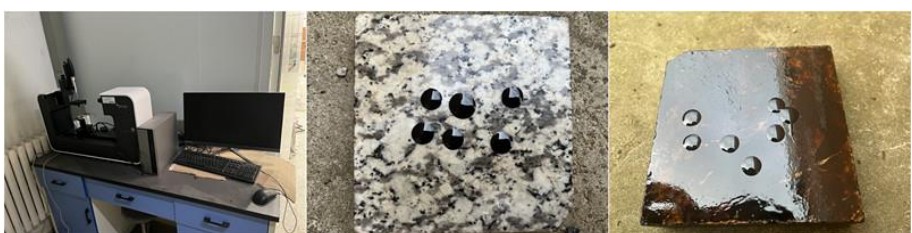

**Figure 9.** Contact angle test.

**Table 10.** Contact angle test results.

| Interface Type | Temperature (°C) | | | Contact Angle (°C) |
|---|---|---|---|---|
| | 155 | 165 | 175 | |
| **Conventional interface** | 30.3 | 27.1 | 23.9 | 27.1 |
| Pre-wet interface | 15.6 | 13.5 | 12.7 | 13.9 |

According to the experimental results in Table 10, it can be seen that after pre-wetting, the contact angle between the asphalt and stone surface becomes significantly smaller, decreasing by 48.7%. The smaller the contact angle, the more compatible the two surfaces are,

and the better the interface properties formed. The results of the contact angle experiment and molecular dynamics simulation are mutually confirmed. The technology of the pre-wet asphalt mixture is feasible.

### 4.5.4. Evaluation of Asphalt–Mineral Interface Adhesion in Dry State

The adhesion work is an indicator of the adhesion of two interfaces, and it is the work required to separate an interface into two interfaces. The greater the adhesion work, the more work needed to separate the interface, the more stable the interface, and the better the interface adhesion. Conversely, the smaller the adhesion work, the more unstable the interface, the less work needed to separate the interface, and the worse the interface adhesion. When the interface between the asphalt and stone is in a dry and anhydrous state, the calculation formula of its adhesion work is shown in Formula (3), and the calculated adhesion work is shown in Table 11.

**Table 11.** Work of adhesion.

| Interface Type | Work of Adhesion (mJ·m$^{-2}$) |
|---|---|
| Un-pre-wet | 36.775 |
| Pre-wet | 38.342 |

It can be seen from the data in Table 11 that when the asphalt forms an interface with the ore, the light components in the asphalt can play a wetting effect, making the asphalt fully contact the surface of the ore, thus forming an interface. In this study, 110# asphalt is used to pre-wetted the surface of the ore, which makes 90# asphalt move faster and coat more evenly when forming the interface with the ore, so as to improve the adhesion work of the asphalt–ore interface. The interface adhesion was improved by 4.3%. Through the calculation of the contact angle and adhesion work of the asphalt on the pre-wetted ore surface, the importance of the pre-wetted ore surface can be clearly seen, and the feasibility of the pre-wetted asphalt mixture technology is determined theoretically.

### 4.5.5. Evaluation of Asphalt–Mineral Interface Adhesion in Water Environment

Water destroys the asphalt–stone interface through a pumping effect under vehicle load. This process is actually the process of breaking one interface and forming two interfaces. When water damage occurs, the energy change in the whole system is called spalling work. The greater the spalling work, the worse the water loss resistance of the asphalt mixture, and the better the water loss resistance. The calculation formula of the spalling work is shown in Formula (10).

$$\Delta W = W_{sl} + W_{al} - W_{sa} \tag{10}$$

where $\Delta W$ is the spalling work, $W_{sl}$ is the stone–water interface energy, $W_{al}$ is the water-asphalt interface energy, and $W_{sa}$ is the stone–asphalt interface energy. The calculation results of the spalling work are shown in Table 12.

**Table 12.** Calculation results of spalling work.

| Interface Type | $W_{sl}$ | $W_{al}$ | $W_{sa}$ | $\Delta W$ |
|---|---|---|---|---|
| Un-pre-wet | 142.192 | 70.132 | 36.775 | 175.549 |
| Pre-wet | 71.529 | 70.132 | 38.342 | 103.319 |

It can be seen from the data in Table 12 that all the peeling work is positive, which indicates that the asphalt mixture will automatically peel from the surface of the ore in the water environment, and this process is spontaneous. If external factors such as vehicle load or environment are added, the peeling process will accelerate. The spalling work of the asphalt mixture using the pre-wetted technology is reduced by 41.1%, which indicates

that the pre-wetted technology can effectively improve the water damage resistance of the asphalt mixture. According to the contact angle test results, it can be seen that the performance of the asphalt mixture using the pre-wet technology has improved significantly, which is consistent with the trend of the molecular dynamics simulation results.

## 5. Conclusions

1.  This research proposes a new method for forming the asphalt–mineral interface, that is, an asphalt mixture pre-wetting technology. This technology facilitates the improvement of the asphalt mixture from the perspective of construction technology. The proposed technology was verified by molecular dynamics simulation software, and the results proved the feasibility of the asphalt mixture pre-wetting technology.
2.  The molecular dynamics simulation results show that the diffusion rate of the asphalt on the surface of the pre-wet mineral material is faster than that on the surface of the non-pre-wet mineral material whether at 25 °C or 165 °C.
3.  The results of the molecular dynamics calculation of the asphalt–mineral interface energy show that the asphalt–mineral interface formed by the asphalt and the pre-wet mineral material has high interface energy, revealing an improved adhesion performance. The pre-wetting effect of the light component facilitates the effective contact and fusion of the asphalt and the surface of the mineral material during the formation of the asphalt–mineral interface, thereby forming a strong interface.
4.  Based on the surface energy theory, an experimental study on the technology of a pre-wetting asphalt mixture is carried out. The experimental results show that the oil–stone interface properties of the pre-wetting asphalt mixture are obviously improved in both a dry state and water environment.

**Author Contributions:** Conceptualization, W.C.; software, Y.C.; formal analysis, Y.C.; investigation, H.L. and Y.W.; data curation, Y.C.; writing—original draft preparation, Y.C.; writing—review and editing, W.C.; project administration, W.C. All authors have read and agreed to the published version of the manuscript.

**Funding:** This research was funded by the National Natural Science Foundation of China (51508223) and the Jilin Province Natural Science Foundation of China (20160101267JC).

**Institutional Review Board Statement:** Not applicable.

**Informed Consent Statement:** Not applicable.

**Data Availability Statement:** All data that support the findings of this study are included within the article.

**Acknowledgments:** The author is very grateful to the National Natural Science Foundation of China (51508223) and the Natural Science Foundation of Jilin Province (20160101267JC) for their strong support for this research.

**Conflicts of Interest:** The authors declare no conflict of interest.

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
