# Peer review of "Study of Asphalt Behavior on Pre-Wet Aggregate Surface Based on Molecular Dynamics Simulation and Surface Energy Theory"

_coatings, doi:10.3390/coatings13101799_

Round 1

Reviewer 1 Report

As the idea of this manuscript was remarkable, its weaknesses in the MD simulation parts force us to REJECT this paper.

The critical results are in the Figure 4 and Table 6 on the interfacial energy calculation of the ‘unpre-wetted’ and ‘pre-wetted’ models. The total energy term of the pre-wetted model was positive, indicating that the simulation is not equilibrated. It is also possible that the simulation time was too short so that the upper asphalt layer had never touched the lower pre-wetting layer and the model cannot emulate the real behavior. (It is advised that author run an NPT simulation with a semi-isotropic box (rescaling only the z-axis) until the box shrinks and the box volume became equilibrated. Then, add the vacuum layer.

There are also some thoughts on the MD methodology for authors to improve their manuscripts before a re-submission.

Aspect ratio of graphical displays in Figures 1 and 4 should be improved.

Calculation methods are not well written and are still hard to follow and reproduce.

- What is the use of showing the technical indicators in Tables 1 and 2? If authors would like to mention about the quality of the 90# and 110# as the base asphalt and the pre-wetting agent, respectively, only the important quantity is needed here. Other detail can be in the supplementary part.

- There is no information about the timestep length (normally 1-2 fs) and the total simulation time

- Figures 1a and 1b: the molecular models were named as saturated “phenol” and aromatic “phenol” but not phenol group was presented in both models. Also, please carefully check the naming of all compounds in Figure 1.

- Initial dimensions of the simulation box also need to be presented.

- Should the density test presented in Table 4 come from the NPT simulations (in which the box volume can be rescaled)?

We also suggested that authors see more example papers on the polymer/surface interactions for simulation protocols and the overall presentation of the manuscript.

Reviewer 2 Report

The study introduces a novel asphalt-mineral interface formation technique called "pre-wet bitumen-mineral," and it employs molecular dynamics simulation and surface energy theory to predict the formation process of this pre-wet asphalt-mineral interface. The following aspects need to be further clarified -

1. Revise the references and add some more relevant citations related to the research. 

2. Methodology section: provide data on modelling and simulation parameters.

3. It is not clear why the authors mentioned the NPT ensemble, initially, but subsequently, utilized the NVT ensemble. Needs clarification.

4.  Why a vacuum layer was considered on the final layer in their study? 

5.  The discussion on diffusion coefficient has been done by referring to Figure 6; whereas, Fig. 6 provides data related to Mean Squared Displacement (MSD). Look into it. 

6. The contact angle has been used for validation. However, the MD simulation results on contact angle should be highlighted to compare with experimental results. 

7. Add a discussion on the validity of the potential function to address the applicability to this specific study.

8.  Add information on the model assumptions, and the choice of parameters, temperatures, and timesteps.  Why were only two temperatures 25°C and 165°C considered for this study?

Improve the grammatical errors. 

Reviewer 3 Report

The study of asphalt behavior on pre-wet aggregate surface based on molecular dynamics simulation and surface energy theory is interesting from scientific point of view.

  1. Can the model presented in the article be used for modified bitumen?

  2. Why were only density and solubility parameters used to verify the rationality of the asphalt molecular model?

  3. Page 11. Missing 'table 7' above table.

Reviewer 4 Report

- how the temperatures of 25 and 165 C were chosen ?

- other experimental tests can also be carried out?

Round 2

Reviewer 1 Report

=========================

Point 1: The critical results are in the Figure 4 and Table 6 on the interfacial energy calculation of the ‘unpre-wetted’ and ‘pre-wetted’ models. The total energy term of the pre-wetted model was positive, indicating that the simulation is not equilibrated. It is also possible that the simulation time was too short so that the upper asphalt layer had never touched the lower pre-wetting layer and the model cannot emulate the real behavior. (It is advised that author run an NPT simulation with a semi-isotropic box (rescaling only the z-axis) until the box shrinks and the box volume became equilibrated. Then, add the vacuum layer.

Response 1: Thanks very much for your valuable comment. Based on the reviewer's suggestions, the author has made changes to the content in 4.2.

Further comment:

-       It is still not clear what are the E_dynamics and E_anneal.

-       Still, poor aspect ratio of Fig 4.2b.

-       The gap seen in Fig 4.2b between two mixture layers is not making sense and cannot represent the system under the atmospheric pressure. It seems like authors did not perform the long enough NPT simulation as suggested.

=========================

Point 4: Calculation methods are not well written and are still hard to follow and reproduce.

Point 6: There is no information about the timestep length (normally 1-2 fs) and the total simulation time

Point 10: We also suggested that authors see more example papers on the polymer/surface interactions for simulation protocols and the overall presentation of the manuscript.

Further comment:

-       The quality of writing in the methodology parts are not much improved. There are still many missing important information, including algorithms for temperature and pressure regulation, treatments of short-range (distance cutoff) and long-range (e.g. PME) interactions, and holonomic constraints

-       Authors seems to misunderstand about the word ‘timestep’, as it looks like 100 ps is the total simulation time.

-       100 ps total simulation time also is way too short for the system to equilibrate (so that two asphalt layers would merge together after a 10-50 ns under an NPT simulation).

=========================

Point 7: Figures 1a and 1b: the molecular models were named as saturated “phenol” and aromatic “phenol” but not phenol group was presented in both models. Also, please carefully check the naming of all compounds in Figure 1.

Response 7: Thanks very much for your valuable comment. Following the recommendations of the reviewers, the authors carefully examined Figure 1, where the model of asphalt molecular components was established according to references 23-26.

Further comment:

-       A phenol should contain a hydroxyl (-OH) group. Therefore, chemical structures in Figs 1a and 1b are still not right.

=========================

There are still more points to discuss regarding the results but it could be of no use as I think the simulation methods are still not reliable enough and the systems are not well equilibrated. Therefore, I have no choice but to reject this manuscript.

Reviewer 3 Report

Thank you for your answers.
